# CDC6 as a Key Inhibitory Regulator of CDK1 Activation Dynamics and the Timing of Mitotic Entry and Progression

**DOI:** 10.3390/biology12060855

**Published:** 2023-06-14

**Authors:** Mohammed El Dika, Damian Dudka, Malgorzata Kloc, Jacek Z. Kubiak

**Affiliations:** 1Department of Biochemistry, Larner College of Medicine, UVM Cancer Center, University of Vermont, Burlington, VT 05405, USA; mohammedeldika@hotmail.com; 2Department of Biology, School of Arts and Sciences, University of Pennsylvania, Philadelphia, PA 19104, USA; dudka@sas.upenn.edu; 3The Houston Methodist Research Institute, Transplant Immunology, Houston, TX 77030, USA; mkloc@houstonmethodist.org; 4Department of Surgery, The Houston Methodist Hospital, Houston, TX 77030, USA; 5Department of Genetics, MD Anderson Cancer Center, The University of Texas, Houston, TX 77030, USA; 6Laboratory of Molecular Oncology and Innovative Therapies, Military Institute of Medicine—National Research Institute (WIM-PIB), Szaserow 128, 04-141 Warsaw, Poland; 7Dynamics and Mechanics of Epithelia Group, Faculty of Medicine, Institute of Genetics and Development of Rennes, University of Rennes, CNRS, UMR 6290, 35043 Rennes, France

**Keywords:** time of mitosis, cell cycle, CDC6, CDK1, cyclins, CDC25, Xic1, mitotic entry

## Abstract

**Simple Summary:**

The kinetics of Cyclin Dependent Kinase 1 (CDK1) activation must be strictly controlled to guarantee a timely and physiological entry into mitosis. CDC6, a known S-phase regulator, has recently been found as a critical component in mitotic CDK1 activation cascade in early embryonic divisions. It acts due to association with Xic1 serving as a *bona fide* CDK1 inhibitor upstream of Aurora A and Polo-Like Kinase 1 (PLK1), both of which are CDK1 activators. Here, we discuss the molecular processes that control the time of mitotic entry focusing on how the CDC6/Xic1 regulatory network affects CDK1 function. We present the existence of two distinct inhibitory mechanisms—Wee1/Myt1- and CDC6/Xic1-dependent—that slow down the kinetics of CDK1 activation, as well as how they interact with mechanisms that activate CDK1. The dynamics of CDK1 activation appear to be modulated by several inhibitors and activators, and their coordinated modulation ensures the robustness and some flexibility of mitotic timing. We therefore suggest a new model that incorporates CDC6/Xic1-dependent inhibition into the CDK1-activation cascade. This enables a better understanding of why cells divide at specific times and how the pathways involved in the timely regulation of cell division are all integrated to finely tune the control of mitotic events.

**Abstract:**

Timely mitosis is critically important for early embryo development. It is regulated by the activity of the conserved protein kinase CDK1. The dynamics of CDK1 activation must be precisely controlled to assure physiologic and timely entry into mitosis. Recently, a known S-phase regulator CDC6 emerged as a key player in mitotic CDK1 activation cascade in early embryonic divisions, operating together with Xic1 as a CDK1 inhibitor upstream of the Aurora A and PLK1, both CDK1 activators. Herein, we review the molecular mechanisms that underlie the control of mitotic timing, with special emphasis on how CDC6/Xic1 function impacts CDK1 regulatory network in the *Xenopus* system. We focus on the presence of two independent mechanisms inhibiting the dynamics of CDK1 activation, namely Wee1/Myt1- and CDC6/Xic1-dependent, and how they cooperate with CDK1-activating mechanisms. As a result, we propose a comprehensive model integrating CDC6/Xic1-dependent inhibition into the CDK1-activation cascade. The physiological dynamics of CDK1 activation appear to be controlled by the system of multiple inhibitors and activators, and their integrated modulation ensures concomitantly both the robustness and certain flexibility of the control of this process. Identification of multiple activators and inhibitors of CDK1 upon M-phase entry allows for a better understanding of why cells divide at a specific time and how the pathways involved in the timely regulation of cell division are all integrated to precisely tune the control of mitotic events.

## 1. Introduction

The entry into mitosis is controlled by CDK1/Cyclin B, also known as MPF (maturation promoting factor). Previous research has shown that cyclin B synthesis is the key for driving the embryonic cell cycle and determining the timing of mitosis in *Xenopus leavis* [1,2,3]. Apparently, it is a key, but it is not the only controlling factor. Cyclin B accumulation during interphase is necessary for CDK1 activation because without cyclin, CDK1 cannot be active as a protein kinase. On the other hand, cyclin degradation through the ubiquitin pathway is necessary for CDK1 inactivation and mitotic exit [1,2,3]. Cyclin B is encoded by several genes and creates a family of B-type cyclins required for mitosis (e.g., cyclins B1, B2, and B3 in human; or cyclins B1-B5 in *Xenopus leavis* [4,5]). De novo B-type cyclin synthesis is required between meiosis I and II during *Xenopus* oocyte maturation [4]; however, despite a steady increase in cyclin B levels in G2 [6], CDK1 activation is biphasic, characterized by a slow phase, followed by a rapid phase attributed to the positive feedback between newly activated CDK1 and its major activating phosphatase CDC25 [6]. How the stable increase in cyclin B level may lead to this biphasic CDK1 activation before the CDK1/CDC25 positive feedback acceleration triggering the final peak of CDK1 activity has been intriguing for the long time. This particularity suggested that the control of CDK1 activation at the initial stages of its mitotic activation might not solely depend on cyclin B accumulation but might also involve an unidentified inhibitor that could counterbalance CDK1 activation, assuring the slow and biphasic mode of this process. Our research group performed a proteomic screen to find novel CDK1 partners in M-phase-arrested *Xenopus leavis* eggs vs. freshly activated ones (5 min post-activation [7]). We showed a rapid 5 min, change in the composition of the CDK1 complex during the period between the MII arrest of oocytes and their activation for development; however, we did not find any classical CDK1 inhibitors, such as INK4 or Cip/Kip, in this screen, which potentially could slow down CDK1 activation [8] in a CDK1 complex. Surprisingly for us, we found a CDC6 protein associated with the active mitotic CDK. CDC6 inhibits CDK1 throughout the whole period of the M-phase (including the highest peak of CDK1 activity just prior its inactivation) [9,10,11,12,13,14]. CDC6 is an evolutionarily conserved member of the AAA + ATPase family and was only known as the S-phase activator [6]. It initiates DNA replication at each origin once per cell cycle to maintain genomic stability and is regulated during the S-phase by another member of the CDK family, namely CDK2. The CDK2/cyclin A-mediated phosphorylation of CDC6 causes its translocation from the nucleus to the cytoplasm, blocking DNA replication [15]. CDK1 can additionally phosphorylate CDC6 to maintain it in the cytoplasm (ibid.). Furthermore, we showed that two proteins, namely CDC6 and Xic1 (the last one is a *bona fide* CDK1 inhibitor identified for its role in later stages of *Xenopus* embryo development), interact within the mitotic CDK1/CDC6 complex. Upon arriving at the peak of the CDK1 activity (measured by histone H1 kinase activity), the Xic1 momentarily dissociates from the mitotic CDK1 allowing for the maximum activation of CDK1 during the M-phase [9]. It is critical to understand how CDC6 and Xic1 receive signals, how they interact, and how they are interdependent; however, the identification of this new inhibitory mechanism towards CDK1 was surprising because it has been well known already for years that CDK1 is inhibited by the posttranslational modification by Wee1/Myt1 kinases. Thus, our data showed that there are two independent mechanisms of CDK1 activity in the game during the mitotic entry.

In parallel to our findings, other researchers continued exploring the molecular mechanisms that accurately regulate the timing of mitosis in *Xenopus laevis* as well as in *C. elegans* and human cells [16,17,18,19,20,21,22,23,24,25]. Importantly, it was shown that CDK1/Cyclin A/B dimers phosphorylate Aurora-A kinase co-factor Bora, and this triggers the activation of the kinase, in turn activating PLK1 kinase and the CDK1’s amplification loop, which helps the cell to enter and progress into M-phase in a timely manner [19,20,21,22,26]. All these data suggested an interplay between inhibitor and activator molecular mechanisms governing mitotic CDK1 activation process, which requires a precise definition of their interrelationship. For this reason, we provide here an overview of the molecular processes underlying timely mitotic entry and progression. As will be discussed in the next paragraphs, there are different mechanisms to control a timely increase in CDK1 activity upon mitotic entry, including the association with other regulators (Xic/CDC6) or modulation of the N-terminal inhibitory phosphorylations. We focus on the CDK1 regulatory network and especially on the function of CDC6/Xic1 and Wee1/Myt1 acting as CDK1 inhibitors. The multiple system of CDK1 activators and inhibitors explains how the mitotic activation of CDK1 is tuned and why cells divide in the strictly defined time frame. We also explain how the various pathways regulating this process are integrated.

## 2. Activation of CDK1 by Association with Cyclins A and B and Its Concomitant Inhibition by Wee1/Myt1 Kinases Post-Translational Modifications

The catalytic region of CDK1 is inaccessible to ATP before association with the cyclin. This is due to the PSTAIR amino acid motifs of CDK1 being positioned on the T-loop, which shuts the entry to the catalytic site. The interaction of CDK1 with cyclins causes the T-loop to change the 3D conformation. This allows other CDK1 regulatory proteins to access specific amino acids. The combination causes a conformational shift of CDK1 in the interior of the ATP binding site due to the rotation of the PSTAIR domain, which is required for kinase activity and facilitates phosphate transfer on the target protein. The substrate specificity of complex CDK1/cyclins is determined by a change in the conformation of the catalytic sites of this kinase [26]. Cyclins are essential for the control of multiple stages of the cell cycle. G1/S cyclins interact with CDK1 during the G1-phase and are essential for S-phase entry and progress. Mitotic cyclins interact with CDK1 and CDK2 during G2 phase and are necessary for mitotic entry. The level of cyclin fluctuates during the cell cycle, whereas the level of CDK1 remains constant throughout. Changes in cyclin levels allow for the periodic activation of CDKs [27,28]. During the G2/M transition, the increased stability of cyclin B mRNA allows for a significant and continuous accumulation of cyclin B protein [28]. This newly synthesized cyclin B associates with CDK1, but due to the inhibitory phosphorylation on CDK1 Thr14 and Tyr15 sites by Wee1 and Myt1 kinases, respectively, the newly formed CDK1/cyclin B complex is catalytically inactive in the cytoplasm [29,30]. Cyclin B will continue to be produced during the G2 phase and increase the amount of CDK1/cyclin B complexes until a critical quantity is reached just before mitotic entry. At this moment, CDK1 kinase is activated by the dephosphorylation of Tyr14 and Thr15 by the CDC25 phosphatase family members (CDC25 A, B, and C), which counterbalance the Wee1/Myt1-dependent inhibition. CDK1 activation also requires the continuous phosphorylation of Thr161 by CAK [31,32,33,34,35,36,37]. During the dephosphorylation process of CDK1 on Tyr14 and Thr15, CDC25B acts as a starter phosphatase for the final pic of the activation of CDK1 [38,39]. Importantly, CDK1/cyclin B phosphorylates and activates CDC25C. This triggers the positive feedback between CDK1 and CDC25 called the MPF activating loop, which is responsible for the final, non-linear CDK1 activation phase [40]. Moreover, active CDK1 phosphorylates and inhibits Wee1/Myt1, thereby eliminating the inhibitory effect of these kinases on CDK1 itself [41,42]. Thus, the amplification loop of CDK1 activation guarantees the extremely rapid mitotic progression following the slow and progressive entry into the M phase.

The beginning of the mitotic CDK1 activation is preceded by the accumulation of CDK1/Cyclin B in the cell nucleus. If cyclin B is not imported to the nucleus, the mitotic entry is halted, which underlies the importance of nuclear localization of the CDK1/cyclin B complex for the mitotic entry [43]; however, the association of CDK1 with cyclin B did not explain how the initiation of CDK1 activation can be triggered. The main problem was that CDK1/cyclin B is efficiently inactivated by Wee1/Myt1 kinases, which did not allow initiation of the kinase activation. The study by Gheghiani et al. (2017) provided the first indications that CDK1/cyclin A may act as a trigger for CDK1 activation [44] while PLK1 activation in late G2 sets up commitment to mitosis. Another observation suggesting an important role of cyclin A in the initiation of the mitotic activation of CDK1, and especially in the control of its timing, was that cyclin A depletion from the *Xenopus* cell-free extract enormously prolonged the time of CDK1 activation [9,20]. The final proof of such a role of cyclin A was shown by Vigneron et al. (2018) who demonstrated that contrary to CDK1/cyclin B, CDK1/cyclin A is not inhibited by Wee1 during interphase. This opened the possibility that accumulation of CDK1/cyclin A may easily attain a critical activity to start the whole pool of cyclin A- and B-associated CDK1 activation. The escape of CDK1/cyclin A from the inhibitory phosphorylation by Wee1/Myt1 was indeed shown to break the post-translational inhibition of CDK1 (ibid.). Additionally, our results showing that cyclin A and cyclin B overexpression in cell-free extracts have different effects on the timing and amplitude of CDK1 activation pointed out that an interplay between CDK1/cyclin A and CDK1/cyclin B is important in the regulation of the timing of CDK1 activation [9]. Indeed, the increase in cyclin A in cell-free extracts modifies only the level of CDK1 activity, while the increase in cyclin B modifies both the level of CDK1 activity and the timing of its full activation, which is accelerated in the latter case (ibid.); however, it must be noted that in this experimental system, the important increase in A- or B-type cyclins may install a competition between the two types of cyclins for the binding with free CDK1 (ibid.).

## 3. Role of PLK1 in CDK1 Activation upon Mitotic Entry

The kinase PLK1 is involved in the MPF amplification loop (being a part of the CDK1 activation process upon M-phase entry) through phosphorylating, and thus activating CDC25C [45,46,47]. PLK1 depletion delays CDK1 activation and stops CDC25C hyperphosphorylation (ibid.). PLK1 docking to CDC25C requires priming, which depends on CDK1-mediated CDC25C phosphorylation [48]. PLK1 phosphorylation of CDC25C triggers its nuclear translocation [49].

As they contain CDK1 kinase in G2 phase, centrosomes are crucial CDK1/Cyclin B regulators [50,51,52]; however, that pool is inactive because CDK1 inhibitors CHK1 and CHK2 kinases are present and/or active [53,54], preventing the pool’s premature activation. The centrosomal pool of PLK1 contributes to CDK1 activation by counteracting the inhibition of the kinase by CHK1 and 2 [52,55]. Injecting anti-PLK1 antibodies into primary human cells or depleting PLK1 from *Xenopus* eggs extracts induces G2 arrest [47,56,57]; however, PLK1-depleted cells enter mitosis in the presence of partially active CDK1, suggesting that partial CDC25C phosphorylation is sufficient for mitotic entry [58]. Additionally, PLK1 regulates the CDK1 activation via phosphorylation of CDK1-inactivating kinase Wee1, which, in turn, promotes its degradation, and thus decreases the inhibitory phosphorylation of CDK1 [41]; however, the discovery that the active CDK1/cyclin A complex promotes PLK1 activation via Bora phosphorylation allowed for a better understanding of the role of Plk1 in CDK1 activation [20], as discussed in the following section. All these findings indicate the function of PLK1 as a key factor for activating CDK1, its downstream activation pathway, and point to the role of CDK1/cyclin A and Bora in this process.

## 4. Role of Aurora A/Bora Complex in CDK1 Activation through PLK1

The serine/threonine kinase Aurora A plays a crucial role in the regulation of mitotic entry [17,18,22,59,60,61]. One of the best-known substrates of Aurora A is kinesin Eg5, whose activity is implicated in centrosome separation and maturation [62,63]. Aurora A induces CDC25B phosphorylation, leading to CDK1/cyclin B activation [64].

It has been shown that Bora is necessary for Aurora A to phosphorylate PLK1 in human cells [65,66]. CDK1-mediated phosphorylation of Bora significantly increases Aurora A’s ability to phosphorylate PLK1 [18,59,60,67], showing the importance of an upstream CDK1 complex for the activation of PLK1. This CDK1 complex has been identified as CDK1/Cyclin A complex [20]. Recently, it has been demonstrated that in G2/M transition, both CDK1/cyclin A and CDK1/cyclin B phosphorylate Bora to induce Aurora A’s phosphorylation of PLK1 and the full activation of the CDK1/cyclin B feedback loop [19,20,24]. More precisely, Aurora A-mediated PLK1 phosphorylation in G2 is enabled by Bora-induced conformational shift in PLK1 kinase. Consequently, PLK1, CDC25, and CDK1/cyclin B complexes form a positive feedback loop promoting entry into mitosis. PLK1 causes the degradation of Bora by the ubiquitin/proteasome pathway [66]. The summary of this regulation is shown in Figure 1.

All these results are consistent with the idea that the Aurora-Bora complex is a crucial component of the CDK1/Cyclin B amplification mechanism and, consequently, of the timely decision-making for mitosis.

## 5. Role of Nuclear Transport in CDK1 Activation and Function

Nuclear translocation is another essential element that controls CDK1 activation. Cyclins A and B1 are differentially located in the cell and undergo cell cycle-dependent nuclear transport in humans [28]. Cyclin A is primarily nuclear from S-phase onwards in both primary human fibroblasts and epithelial carcinoma cells. Cyclin A is associated with condensing chromosomes during prophase but not with condensed chromosomes during metaphase. Cyclin B1, on the other hand, accumulates in the cytoplasm of interphase cells and enters the nucleus only at the start of mitosis before the nuclear lamina breaks down. In human cells, cyclin B1 binds with condensed chromosomes in prophase and metaphase, as well as the mitotic apparatus [28]. In *Xenopus,* cyclin B1 nuclear localization is regulated by phosphorylation [43,68]. Five phosphorylation sites have been identified: Ser2, Ser94, Ser96, Ser101, and Ser113 [68,69]. Four of these sites are inside the cytoplasmic retention signal (CRS) domain, a region of 78–127 residues in *Xenopus* cyclin B1 that was previously assumed to be required for cytoplasmic retention [70]. When these Ser residues are changed to Ala, cyclin B1 loses its capacity to translocate to the nucleus resulting in reduced quantities of it during oocyte maturation. Incorporation of a nuclear localization signal (NLS) into the Ala mutant restores the biological activity to induce oocyte maturation, implying that cyclin B1 phosphorylation within the CRS is required for nuclear translocation [43]. This is supported by the fact that changing these Ser residues to Glu causes cyclin B1 accumulation in the nucleus and accelerates oocyte maturation (ibid.). In *Xenopus*, cyclin B1 translocation to the nucleus appears to also be essential for mitotic entry. Mutations that inhibit cyclin B1’s nuclear localization block its mitotic functions, but mutations that result in constitutive nuclear localization of cyclin B1 are sufficient to trigger premature mitotic processes under certain conditions [43,71,72].

Changes in subcellular localization of cyclin B1 are known to impact/drive nuclear translocation and activation of CDK1. Cyclin B1 has two domains that ensure its cytoplasmic localization in G2 phase: a nuclear export domain NES (nuclear export signal) and a cytoplasmic retention domain (CRS) [72]. These cyclin B1 domains are phosphorylated by CDK1 and PLK1 in mitosis [73,74]. Disruption of cyclin B1 NES phosphorylation prevents the cyclin from being retained in the nucleus [52,75]. Its nuclear import is also triggered by associating with cyclin F [76] and β-importin [77,78]. Once in the nucleus, CDK1/cyclin B1 phosphorylates a wide range of substrates, including the retinoblastoma protein (pRb), an important tumor suppressor gene. pRb disassembles the histone deacetylase complex (HDAC) and releases transcription factors E2F-1 and DP-1, triggering the transcription of genes whose products are required for S-phase progression, including cyclin A, cyclin E and CDC25 [79,80,81]. Additionally, vimentin, nuclear lamins, microtubules and other cytoskeleton proteins are among key CDK1 targets that govern mitotic progression [40,82,83,84].

## 6. CDC6 as an Upstream Regulator of CDK1 through Its Inhibition

CDC6 is essential for the folding, unfolding, and degradation of proteins. Its 200–250 amino acid ATPase region is necessary for the assembly of pre-RCs. More specifically, CDC6 attaches to ORI (ORC-attached replication origins) on chromosomes to generate an ORI-CDC6-ORC complex, which then binds Cdt1 to recruit several MCM2-7 helicase subunits to ORI. This is powered by its ATP-hydrolase activity [85,86,87,88]. Furthermore, CDC6 has a phosphorylation site for PLK1 in its N-terminal side, and three consensus sites of phosphorylation for CDK1 and CDK2 [89]. CDC6 has a leucine zipper domain for protein–protein interaction and Cy-motif for cyclin interaction (ibid.). It also has amino-acid sequences of D- and KEN-box motifs which are necessary for ubiquitination by the anaphase-promoting complex/cyclosome (APC/C), the main mitotic ubiquitin ligase, which participates in CDC6 degradation by the proteasome in the G1/G0 phase (ibid.). It was shown that in yeast, CDC6 acts as a CDK1 inhibitor during the mitotic exit and that it inhibits CDK1-dependent histone H1 phosphorylation in vitro [90]. Moreover, together with SIC1 (Stoichiometric inhibitor of Cdk1-Clb) (an inhibitor of CDK1) and CDH1 (Cdc20 homolog 1), an activator of the APC (anaphase promoting complex), CDC6 participates in the activation of APC necessary for cyclin B degradation and mitotic exit [91]. Both SIC1 and CDC6 have been shown to associate with CDK1/cyclin B complexes with low nanomolar affinity, which is partially facilitated by the docking of the phospho-adaptor CKS1 [92,93]. The deletion of the CDC6 N-terminal CDK1-binding site increases CDK1 activity in mitosis, indicating that the CDC6 N-terminus is critical for the regulation of CDK1 activity [91]. The depletion of CDC6 causes a delay in mitotic exit. This finding was further supported by an in vitro assay showing the CDC6-dependent inhibition of CDK1 kinase activity mediated by a CDK-specific cyclin docking motif, LxF, in CDC6 and the phospho-adaptor Cks1, leading to shielding of the degron and CDC6 sequestration by CDK during mitotic exit [92].

In human cells, CDC6 also associates with and inhibits CDK1 at mitotic exit [94]. In mitosis, CDC6 is hyperphosphorylated in correlation with an increase in the level of PLK1; however, CDC6 is hypophosphorylated in PLK1-depleted cells, even when cyclins A and B are present at high levels. PLK1 phosphorylates CDC6 during mitosis, and this promotes its binding to CDK1, which in turn downregulates CDK1 activity. This results in the activation of separase, an enzyme that resolves sister chromatid cohesion during the metaphase-to-anaphase transition, leading to the mitotic exit. CDC6 depletion results in defects in chromosome segregation and cytokinesis, leading to aneuploid cells with increased CDK1 activity [94].

In addition to its role in mitotic exit in yeast and human cells [90,95,96], our group has also showed the role of CDC6 in delaying mitotic entry by inhibiting CDK1 activity in *Xenopus laevis* premitotic cell-free extracts [9,10,11,12,13,14]. CDC6 associated with CDK1 during the M-phase rapidly dissociates from this kinase immediately after CDK1 inactivation, suggesting that once CDK1 is fully inactivated by separation from cyclin B, the association with CDC6 becomes unnecessary. It was a surprising observation in the light of the knowledge at that time that there are suggestions of a potential role of CDC6 only in the mitotic exit, and not entry and progression [9]. The important hint to the role of CDC6 in CDK1 activation process was its abovementioned function in the M-phase exit described in *yeast* and human cells. Through biochemical analysis of *Xenopus* embryo cell-free extracts we showed that a recombinant CDC6 protein acts as an inhibitor of CDK1 during the first embryonic mitosis in *Xenopus leavis* [9,10,11,12,13,14]. Importantly, when endogenous CDC6 is depleted, the first, initial and slow phase of CDK1 activation is removed, and the kinase activation ceases to be biphasic. This, in turn, accelerates the timing of mitosis and the pic of CDK1 activity induced by the CDC25/CDK1 activation loop [9,10,11,12,13,14]. Adding a recombinant CDC6 completely reverses all these effects [14,16,95]. Additionally, exogenous cyclin A or cyclin B added to *Xenopus laevis* cycling extracts leads to distinct regulation of the kinetics of this mitotic kinase activity [9]. We found that while adding cyclin A increases only the level of CDK1 activity, adding cyclin B increases the level of CDK1 activity and prolongs the timing of its activation. On the other hand, the absence of cyclin A causes an important delay in CDK1 activation. While adding recombinant CDC6 to cyclin A-depleted extract does not affect the timing of the delayed mitosis; however, it does inhibit the CDK1 activity. Thus, it demonstrates that the CDC6-dependent mechanism inhibits both CDK1/cyclin A and CDK1/cyclin B (ibid). This implies in turn that a CDC6-dependent mechanism of CDK1 inhibition plays a role in also delaying the initial activation of CDK1/cyclin A.

In addition to this inhibitory function of CDC6 regarding CDK1 activity, the switch from cyclin A- to cyclin B-dependent CDK1 activity at the beginning of the M-phase may also contribute to the biphasic CDK1 activation pattern during the M-phase entry. Our mathematical model based on mass action kinetics shows a diauxic character of the CDK1 activity growth and allows for the simulation of changes in the dynamics of CDK1 activation in relation to changing concentrations in CDC6 and the dynamics of CDK1/cyclin B/CDC6 complexes formation [13]. The diauxic character of the process of CDK1 activation is strictly related to the presence of at least three inflection points in the curve of CDK1 activation. The two-step mode of CDK1 activation in *Xenopus laevis* cell-free extract resembles the dynamics of the diauxic growth of bacterial or yeast population upon the switch from one sugar to another one when the first becomes exhausted in the culture medium [13,97]. It suggested an analogic change concerning CDK1 activity, and the switch between cyclin A and cyclin B as a partner of CDK1 during the initial phases of CDK1 activation fits perfectly with this view.

In mouse zygotes, the depletion of CDC6 also causes acceleration of the entry into mitosis, similarly to the *Xenopus* cell-free extract. It showed the physiological relevance of CDC6 not only in the cell-free system but also in intact cells [11,14]. In addition, CDC6 regulates both meiotic entry and the metaphase-to-anaphase transition during the first meiotic division in mouse oocytes [98].

Taken together, these results show that CDC6 is involved in a CDK1 inhibitory mechanism both in vitro and in vivo during mitotic (and meiotic) entry, progression and exit.

## 7. Regulation of CDC6 by PP2A

The serine/threonine phosphatase PP2A is the major protein phosphatase involved in dephosphorylating CDK substrates [99]. This PP2A function is well-conserved across eukaryotes, including yeast and humans [100,101]. The majority of CDK1 substrates are dephosphorylated by PP2A [101,102,103,104,105]. Inhibiting PP2A using okadaic acid triggers premature CDK1 activation in human cells and those of *Xenopus*, mouse, starfish, and other organisms [106,107,108,109,110,111,112].

In yeast, the PP2A heterotrimeric complex is composed of two catalytic C subunits (Pph21 and Pph22), a regulatory B subunit (Cdc55, Rts1, or Rts3), and a scaffold A subunit (Tpd3) [113,114,115]. Among these subunits, the regulatory B subunit is responsible both for the cellular localization and PP2A substrate specificity [116]. Cdc55 (also known as B55 in mammals) and Rts1 have both been involved in mitotic progression. Cytoplasmic PP2A/Cdc55 dephosphorylates and inhibits the CDK1 inhibitor Swe1 [117,118], while nuclear PP2A/Cdc55 prevents the anaphase onset by dephosphorylating Cdc20, a co-factor of APC/C [119,120,121,122]. Furthermore, PP2A/Cdc55 also dephosphorylates Net1, an inhibitor of CDC14 phosphatase [123]. CDC14 release from the nucleolus during early anaphase is mediated by the FEAR and MEN networks and results in mitotic exit through Net1 dephosphorylation [124,125,126,127].

Recently, it has been demonstrated that CDC6 is dephosphorylated by PP2A/Cdc14phosphatases in yeast, which triggers the mitotic exit. PP2A/Cdc55 and CDC14 directly dephosphorylate CDC6 at different CDK1 sites, causing CDC6 stabilization [128,129]. These findings support a scenario in which CDC6 dephosphorylation is required to abolish the inhibition of CDK1 at the mitotic exit.

## 8. Role of PP2A-Greatwall/Mastl Pathway in Potential CDC6 Regulation

Greatwall/Mastl kinase is a conserved regulator of mitotic entry through PP2A phosphatase regulation [130]. There was no evidence from previous studies that PP2A-Greatwall/Mastl was part of a regulatory network during the initial stages of mitotic entry; however, it is well established now that Greatwall/Mastl kinase triggers the activation of CDK1 and other cell cycle regulators preparing the cells for mitosis [130,131,132,133,134,135,136]. Upon activation, Greatwall/Mastl takes part in the amplification cycle downstream of CDK1/Cyclin [137]. It has been shown that Greatwall/Mastl controls PP2A activity in *Xenopus* egg extracts [101,102,132]. Greatwall/Mastl phosphorylates the thermostable protein Arpp-19 (cAMP-regulated phosphoprotein-19) and endosulfine protein (Ensa) to inhibit the activity of PP2A-B55 and triggers mitotic entry [101,130,136]. See Figure 1 for details.

These findings suggest an interaction between CDC6-dependent machinery and Greatwall/Mastl kinase activity. Does CDC6 inhibit Greatwall/Mastl and its substrates to delay mitosis, or is CDC6 involved in the activation of PP2A directly? Addressing these questions is of great relevance to highlight the CDC6 regulatory mechanism necessary for mitotic regulation.

## 9. A Model of CDC6 Involvement in CDK1 Activation

Beyond a simple mechanism of accumulation and degradation of cyclins A and B, the control of cell division is orchestrated by a complex network of regulators that include kinases and phosphatases, as well as inhibitory partners. They are required to precisely regulate CDK1 levels determining the timing of mitotic events and the timely progression of mitosis. CDC6 participates in the inhibitory mechanisms controlling CDK1/cyclin A and CDK1/cyclin B activities and regulates the time of mitosis in *Xenopus laevis* and mouse embryos [9,11]. This is corroborated by the data obtained in yeast and human cells [90,91,92,93,94]. This regulatory mechanism is particularly important for the coordinated activation of mitotic CDK1 and the control of both the timing and amplitude of CDK1 kinase activity during early embryo cleavage divisions, which are highly synchronous in *Xenopus*; therefore, this regulation is required for the coordination between cell divisions and the embryo genetic developmental program.

According to recent research [19,20,22,24], CDK1/cyclin A-B phosphorylates Bora to induce Aurora A’s phosphorylation of PLK1 to fully activate the CDK1/cyclin B amplification loop. This shows that both CDC6 and Aurora-Bora networks are crucial components of the CDK1/Cyclin B amplification mechanism and, consequently, of the control of the timing of mitosis. Here, we present a new model for the entry into mitosis, integrating CDK1, CDC6/Xic1, Wee1/Myt1 and Aurora A-Bora-PLK1 network that, together, regulate the activation of the CDK1 amplification loop (Figure 2).

During interphase, cyclin A accumulates and binds to CDK1. Judging by our experiments [9], we postulate that the initial pool of CDK1/cyclin A, which escapes from Wee1/Myt1 inhibition, is under an inhibitory control of the CDC6-dependent mechanism. Thus, CDK1/cyclin A complex is inhibited only by CDC6 and not by Wee1/Myt1 kinases, in contrast to the CDK1/cyclin B complex. Thus, the first active CDK1/cyclin A molecules appear when they escape from CDC6-dependent inhibition, while CDK1/cyclin B complexes are inhibited by both CDC6/Xic1 and Wee1/Myt1. This single inhibition of the CDK1/cyclin A pool and the double inhibition of the CDK1/cyclin B pool shows the major role of CDC6 in the control of the timing of this event. The levels of cyclin A and B continue to rise throughout time and newly synthesized cyclin molecules become progressively associated with CDK1. The active CDK1/Cyclin A and the CDK1/Cyclin B become more numerous, and they can phosphorylate their substrates, such as Bora and Greatwall/Mastl, also known as Gw1 in *Xenopus*. During this phase, CDK1 activity is in its slow phase of increase. Bora interacts with PLK1 and Aurora kinase A, causing PLK1 to be activated. While part of the amplification loop, PLK1 also allows for the controlled activation of CDK1 by promoting CDC6-mediated inhibition of CDK1, leading to a gradual CDK1 activity increase [11,94]. Gw1 phosphorylation protects CDC25 from dephosphorylating, enabling it to remain active and effectively activate CDK1 activity. During that period, cyclins accumulate concurrently and associate with CDK1 to produce new active CDK1 molecules. This results in a constant increase in CDK1 activity level, overcoming the CDC6/Xic1-inhibition mechanism. At that time, CDK1 activation enters the rapid phase of activation. When this occurs, CDK1 action becomes efficient in activating CDC25, and CDC25 phosphorylation continues to be further activated over time. At that point, the CDC25 and CDK1 amplification loop starts functioning efficiently, which causes the extremely rapid and massive CDK1 activation; therefore, the full activation of the CDK1 amplification loop occurs, and mitosis progresses.

## 10. Conclusions

CDC6 plays an important inhibitory role in CDK1 activation dynamics and the timing of mitotic entry and progression. it is essential to fully investigate the molecular mechanisms by which CDC6/Xic1 exerts its mitotic function. Upon the association of CDC6, the active site of CDK1 could undergo conformational changes in its PSTAIRE helix and T-loop. This type of conformational shift could inhibit the enzyme from properly interacting with ATP and inactivates the kinase. In yeast, the CDC6 inhibition is mediated by an M-phase-cyclin-binding motif [95]. As a result, it is critical to investigate whether CDC6 inhibits CDK1 in all species, including mammals, using the same inhibition mechanism. A deeper understanding of CDC6 interactions with other mitotic factors will clarify its role in mitosis and in the regulation of the mitotic timing. It may also offer further evidence for multiple roles that CDC6 plays in regulating the cell cycle and embryonic developmental program.

## Figures and Tables

**Figure 1 biology-12-00855-f001:**
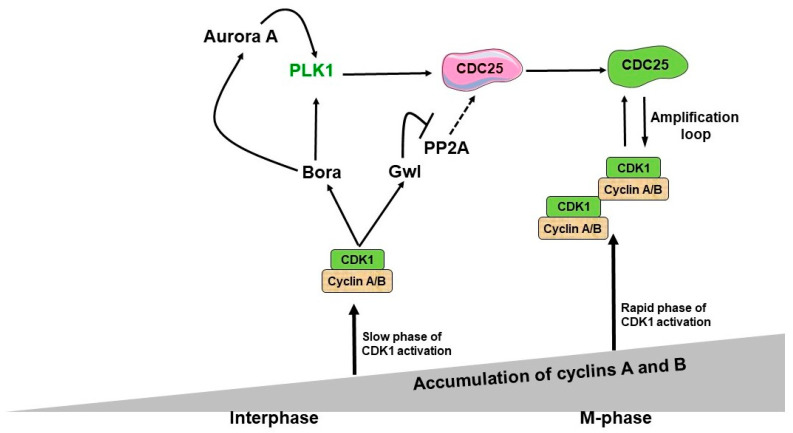
A regulatory network between CDK1, Bora-Aurora A, Plk1, Gw1-PP2A, and CDC25 upon mitotic entry. Before mitosis, CDK1 is activated when cyclins A and B reach a threshold concentration. The mitotic cyclin-associated CDK1 phosphorylates the Bora protein, activating Plk1. Additionally, Bora and Aurora A kinase interact to stimulate Plk1 activity. Parallel to this, CDK1 stimulates Gw1 activity, which inhibits PP2A (Protein Phosphatase 2A) and promotes the activation of CDC25. This activates CDK1’s amplification loop, which blocks PP2A via Gw1 and effectively activates Plk1 by Bora-Aurora A. As a result, the regulatory network between all these proteins controls the time when CDK1 amplification becomes fully activated, and how the mitosis progresses after that.

**Figure 2 biology-12-00855-f002:**
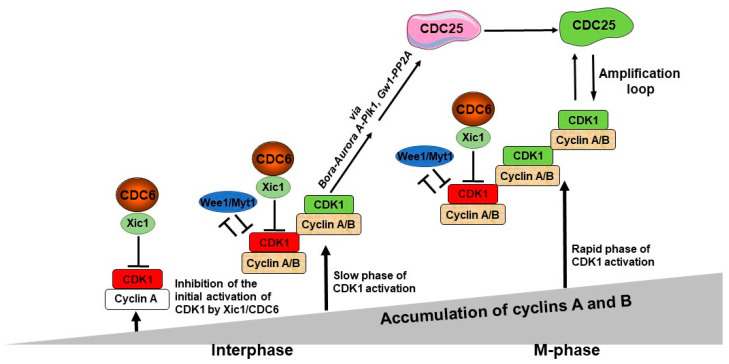
A model of the role of CDC6 in the control of the M-phase entry. The first, potentially active CDK1 molecules start to appear after the cyclin A level reaches a predetermined threshold, but they are actively inhibited by CDC6/Xic1. The first active CDK1/cyclin A molecules appear when they escape from CDC6-dependent inhibition, while CDK1/cyclin B complexes are inhibited by both CDC6/Xic1- and Wee1/Myt1-dependent mechanisms. This is the step of the initial activation of CDK1. In parallel, both cyclin A and B levels continue to increase with time and keep associating with CDK1. The active CDK1/cyclin A and CDK1/cyclin B complexes become more abundant, and they, or only CDK1/cyclin B, become able to phosphorylate its substrates—Bora and Gw1. Due to this activation of Plk1 via these two distinct pathways, the procedure is highly effective, which in turn promotes CDC25 activity acting as the CDK1 amplification loop.

## Data Availability

Not applicable.

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
