# Peer review of "CDC6 as a Key Inhibitory Regulator of CDK1 Activation Dynamics and the Timing of Mitotic Entry and Progression"

_biology, 2023, doi:10.3390/biology12060855_

Round 1

Reviewer 1 Report

Report

Entry into mitosis is a central checkpoint in most types of the eukaryotic cell cycle. In general entry into mitosis is controlled by activation of Cdk1. Multiple conserved mechanisms are  involved, notably T14Y15 dephosphorylation by Cdc25 and complex formation with mitotic cyclins. Simultaneously direct and indirect inhibitors are inactivated such as DNA checkpoint and inhibitory proteins such as Sic1. Beside these highly conserved mechanisms, additional factors impinge on Cdk1 activation depending on cell type, mode of the cell cycle, species and outside conditions. The review focuses a function of CDC6 in mitotic entry and Cdk1 activation. Cdc6 is a well characterized factor for initiation of DNA replication preventing a second start of replication at a given ORI. The authors discuss recent reports about an association of CDC6 with mitotic Cdk complexes and functional analysis. The authors compare this CDC6 function to other pathways of mitotic control. Overall the review presents an accessible overview of the literature and discusses conceivable modes of action. 

Although specialized on a single factor, I would expect the review to be useful for a wide audience. 

Before moving to publication the authors may consider the following comments which will improve accessibility and readability of the article.

-       The authors may revise the figure. The current version is too dense and overcrowded. In this way it will not be useful. I would recommend to split into two or three smaller figures each for single aspects. 

-       The text is written in a very generalized way, although many data are based on studies with Xenopus extracts or embryos. I would recommend to clearly state this also in the abstract. In other cell cycle modes or species, the changes in cyclin levels are not a limiting factor, for example. 

-        

Minor comments

L 18.–34:  mention in the abstract that the review relates mostly to the Xenopus system

L 18  “.. for early embryo…”  delete “the”

L 19  “ubiquitously conserved” conserved proteins are found in many species. Beter write “ .. the conserved protein kinase CDK1…”

L 21 delete article   “.. in mitotic CDK1…

L 24  delete article “… of mitotic timing..

L 29 to controlled by a system of multiple activators and inhbitors…..

L 37  The introduction should provide a brief overview in one paragraph. Details will be presented and discussed in the following paragraphs. The current paragraphs (L38–108) are too long and go already in details, which will be repeated later on. 

L 53, 67  “pic”  meaning to clear to me (peak??)

L54 (ibid.)  meaning no clear to me

L 124  substitute “explosive” with “non-linear”

L215 “the lack maturation”  something missing

L262   “a stoichiometric inhibitor” not clear what it means

L284  better … in delaying mitotic entry by inhibiting CDK1…

L318–340:  First introduce the well established functions of CDC6 in DNA replication. Only then review data about mitosis control.

Author Response

Responses to Referee #1.

  The authors may revise the figure. The current version is too dense and overcrowded. In this way it will not be useful. I would recommend to split into two or three smaller figures each for single aspects. The original figure is divided into two separate ones to make it easier to follow the schematic.

-       The text is written in a very generalized way, although many data are based on studies with Xenopus extracts or embryos. I would recommend clearly state this also in the abstract. In other cell cycle modes or species, the changes in cyclin levels are not a limiting factor, for example. Corrected

-        

Minor comments

L 18.–34:  mention in the abstract that the review relates mostly to the Xenopus system. Done

L 18  “.. for early embryo…”  delete “the” . Corrected

L 19  “ubiquitously conserved” conserved proteins are found in many species. Beter write “ .. the conserved protein kinase CDK1…”. Corrected

L 21 delete article   “.. in mitotic CDK1…. Corrected

L 24  delete article “… of mitotic timing... Corrected

L 29 to controlled by a system of multiple activators and inhbitors…... Corrected

L 37  The introduction should provide a brief overview in one paragraph. Details will be presented and discussed in the following paragraphs. The current paragraphs (L38–108) are too long and go already in details, which will be repeated later on. Done

L 53, 67  “pic”  meaning to clear to me (peak??). Corrected

L54 (ibid.)  meaning no clear to me. Ibid. means in the same source (used to save space in textual references to a quoted work which has been mentioned in a previous reference).

L 124  substitute “explosive” with “non-linear”. . Corrected

L215 “the lack maturation”  something missing. Corrected

L262   “a stoichiometric inhibitor” not clear what it means. Corrected

L284  better … in delaying mitotic entry by inhibiting CDK1. Corrected

L318–340:  First introduce the well established functions of CDC6 in DNA replication. Only then review data about mitosis control. Done.

Reviewer 2 Report

El Dika et al summarize our current knowledge of Cdk1 activation upon mitotic entry. They focus on explaining the role of CDC6 is delaying Cdk1 activation during the first, long phase of cyclin accumulation and explain how then release of inhibition contributes to the second, rapid phase of activation. Beyond their focus, the authors give detailed information on Cdk1 activation collecting observation from yeast, frogs and men that result in a consistent picture of molecular insights underlying M-phase entry.

Taken together, this is a very useful and well-written review that helps the community to keep track of the current knowledge or to allow non-experts to rapidly gain a comprehensive idea of this process.

I am therefore in favor of publication of the review in biology while suggesting some issues that may be taken care of prior to publication.

General / major:

The intro clarifies the ideas. It may benefit from stating upfront that we are talking about different mechanisms to generate the rapid increase in CDK1 activity upon mitotic entry, as it is worked out in the next chapters: by association with other regulators (Xic/CDC6) or by directly or indirectly controlling the N-terminal inhibitory phosphorylations.

I appreciate the figure that summarizes the entire network in the end. However, a couple of figures more showing smaller entities (e.g. the different roles of PLK1 for mitotic entry) would help to keep track of what is explained in the text.

Even though most readers will be experts in the field, it may be worth introducing the concept of cdk activation via cyclins and their timely production / degradation to control cell cycle transitions

The detailed introduction to CDC6 comes only late; its functions in the regulation of replication during S-phase should be emphasized briefly when the protein is initially introduced.

The connection of CDC6 and Xic1 is not completely clarified. Which molecule is doing what exactly to inhibit cdk1, how do they receive signals, are they interdependent, how do these two interact with each other? 

the overall logic, nicely developed in the introduction, is not explicitly evident in the abstract.

Minor:

Some spelling and grammatical errors remain; these should be carefully checked

Is pic also peak?

I don't understand the description of inhibiting nuclear export of cyclin B. Phosphorylation masks the NES of cyclinB which is shown in Ref. 79. The authors write “Once in the nucleus, CDK1 binds the protein CRM1 (Chromosome Region Maintenance 1), which, in turn, prevents the export of cyclin B from the nucleus [79].“ This should be clarified.

The authors have developed a mathematical model of the diauxic character of CDC6s’ action. They only mention this in one sentence (lanes 308/9) but do not remind us about the significance of their studies. What do we learn from the model that was not exactly worked out experimentally before?

Lane 406: PLK1 is activated via these two mechanisms; only one (activation via Bora/Aurora A) is explained.

Lanes 409/10 “…allowing it to efficiently activate cdk1” is oddly misleading

Some spelling and grammatical errors remain; these should be carefully checked

Author Response

Responses to Referee #2.

 General / major:

The intro clarifies the ideas. It may benefit from stating upfront that we are talking about different mechanisms to generate the rapid increase in CDK1 activity upon mitotic entry, as it is worked out in the next chapters: by association with other regulators (Xic/CDC6) or by directly or indirectly controlling the N-terminal inhibitory phosphorylations. The introduction is shortened.

I appreciate the figure that summarizes the entire network in the end. However, a couple of figures more showing smaller entities (e.g. the different roles of PLK1 for mitotic entry) would help to keep track of what is explained in the text. Done

Even though most readers will be experts in the field, it may be worth introducing the concept of cdk activation via cyclins and their timely production / degradation to control cell cycle transitions. Developed in paragraph 2

The detailed introduction to CDC6 comes only late; its functions in the regulation of replication during S-phase should be emphasized briefly when the protein is initially introduced. Done

The connection of CDC6 and Xic1 is not completely clarified. Which molecule is doing what exactly to inhibit cdk1, how do they receive signals, are they interdependent, how do these two interact with each other? Yes, these are unknown inquiries, and they must be addressed in future study.

the overall logic, nicely developed in the introduction, is not explicitly evident in the abstract. Corrected

Minor:

Some spelling and grammatical errors remain; these should be carefully checked

Is pic also peak? . Corrected

I don't understand the description of inhibiting nuclear export of cyclin B. Phosphorylation masks the NES of cyclinB which is shown in Ref. 79. The authors write “Once in the nucleus, CDK1 binds the protein CRM1 (Chromosome Region Maintenance 1), which, in turn, prevents the export of cyclin B from the nucleus [79].“ This should be clarified. . It was removed

The authors have developed a mathematical model of the diauxic character of CDC6s’ action. They only mention this in one sentence (lanes 308/9) but do not remind us about the significance of their studies. What do we learn from the model that was not exactly worked out experimentally before?  Our mathematical model based on mass action kinetics shows a diauxic character of the CDK1 activity growth and allows to simulate changes in the dynamics of CDK1 activation in relation to changing concentrations in CDC6 and the dynamics of CDK1/cyclin B/CDC6 complexes formation [20]. The diauxic character of the process of CDK1 activation is strictly related to the presence of at least three inflection points in the curve of CDK1 acti-vation. The two-step mode of CDK1 activation in Xenopus laevis cell-free extract resembles the dynamics of the diauxic growth of bacterial or yeast population upon the switch from one sugar to another one when the first becomes exhausted in the culture medium [20,98]. It suggested an analogic change concerning CDK1 activity, and the switch between cyclin A and cyclin B as a partner of CDK1 during the initial phases of CDK1 activation fits perfectly with this view.

Lane 406: PLK1 is activated via these two mechanisms; only one (activation via Bora/Aurora A) is explained. Corrected

Lanes 409/10 “…allowing it to efficiently activate cdk1” is oddly misleading. . Corrected

Reviewer 3 Report

In this review El Dika et al., the authors present a general review of the main signaling pathways that regulate entry, progression and mitosis exit. As the authors work on the role of the protein cdc6, this protein has a central place in this review. This is not a problem, as it gives a different view of the question of mitotic entry. After an introduction, the authors discuss the role of the Cdk1-Cyclin A/B complex in inhibiting the Myt1/Wee1 protein kinase, how the Plk1 protein kinase regulates this signalling pathway and how the Aurora-A/Bora/Cdk1 Cyclin A complex acts to activate Plk1.
The authors also discuss the importance of subcellular localization of Cdk1-Cyclin B complexes and the inhibitory role of Cdc6 protein on the Cdk1-Cyclin B complex.
Control of entry into mitosis is not only regulated by the Cdk1-Cyclin A/B, Plk1 and Aurora-A protein kinases, but also by the PP2A-B55 protein phosphatase. This protein phosphatase is itself negatively regulated by the Greatwall/Mastl protein kinase. This kinase, conserved from yeast to man, does not directly inhibit PP2A-B55, but phosphorylates two proteins called Arpp19 and Ensa, which once phosphorylated, bind and inhibit the activity of the PP2A-B55 complex. Greatwall/Mastl is essential for proper entry and progression through mitosis. Interestingly, PP2A B55 regulates Cdc6 by dephosphorylation. Finally, the authors conclude this review with a picture that neatly sums up the complexity of the molecular mechanisms involved in cell entry into mitosis.
The ms is well organized with significant information, however, I have some requests before final acceptation:

1. In the abstract, there are a few words mistakenly written with hyphens.
2. In paragraph 2, which describes how Wee1 is degraded, it would be interesting to mention the work of Li F et al.... d'Andrea AD) Mol Cell. 2020. which describes an interesting regulation of Wee1 protein stability in a DNA damage checkpoint condition.
3. It would be more comprehensible to specify which PP2A complex dephosphorylates the Cdc6 protein. This is PP2A-B55, which we understand to be PP2A-Cdc55.
4.Mastl in humans/mice is the counterpart of Greatwall in Xenopus/drosophila. It is therefore preferable to write Greatwall/Mastl

The manuscript is well-written

Author Response

Responses to Referee #3.

  1. In the abstract, there are a few words mistakenly written with hyphens. . . Corrected
    2. In paragraph 2, which describes how Wee1 is degraded, it would be interesting to mention the work of Li F et al.... d'Andrea AD) Mol Cell. 2020. which describes an interesting regulation of Wee1 protein stability in a DNA damage checkpoint condition. References added
    3. It would be more comprehensible to specify which PP2A complex dephosphorylates the Cdc6 protein. This is PP2A-B55, which we understand to be PP2A-Cdc55. Cdc55 (also known as B55 in mammals)
    4.Mastl in humans/mice is the counterpart of Greatwall in Xenopus/drosophila. It is therefore preferable to write Greatwall/Mastl. Corrected